

,

# Sea Ice Freeboard Extrapolation from ICESat-2 to Sentinel-1

Karl Kortum[1], Suman Singha[2, 3], and Gunnar Spreen[4]

[1]Remote Sensing Technology Institute, German Aerospace Center (DLR), Bremen, Germany
[2]National Center for Climate Research, Danish Meteorological Institute (DMI), Copenhagen, Denmark
[3]Department of Geography, University of Calgary, Calgary, Canada
[4]Institute of Environmental Physics, University of Bremen, Bremen, Germany

**Correspondence:** Karl Kortum (karl.kortum@dlr.de)

**Abstract.** The ICESat-2 laser altimeter can capture sea ice freeboard along track at both high vertical and high spatial resolution. The measurement occurs along three strong and three weak parallel beams. Thus the across track-direction is only very sparsely covered and capturing the two-dimensional spatial distribution of freeboard at high resolution by this instrument alone is not possible. This work shows how in early Arctic Winter (October, November) Sentinel-1 synthetic aperture radar (SAR) acquisitions can help bridge this gap and meaningfully extrapolate the freeboard measurements to a full two-dimensional mapping. To achieve this it is sufficient to use the SAR HV backscatter to sort the pixels by intensity and then map freeboards measured from altimetry in the area via the cumulative distribution functions. With the presented algorithm, snow and ice freeboard derived from altimetry can be meaningfully extrapolated to Sentinel-1 SAR acquisitions, unlocking an extra dimension of Arctic freeboard monitoring at high spatial resolution, with errors between 10.5 cm and 6 cm for resolutions between 100 m and 400 m.

## 1 Introduction

Due to prevalent feedback loops amplification in the Arctic makes it Earths most affected region by climate change (Serreze and Barry (2011); Wendisch et al. (2023) present thorough overviews of the observed amplification). Along with its critical role in Earth's response to the global warming, it is also one of the hardest places to monitor consistently. The environment's remoteness and hostility to the human organism makes in-situ measurements difficult to obtain. As a result, the global community relies on remote sensing for observing change in the polar regions in a continuous manner. Space-borne photography in the optical spectrum is only feasible during polar day for approximately half a year. Passive microwave and other active remote sensing techniques thus move to the forefront of operational monitoring of the polar regions. Passive microwave instruments and corresponding retrieval algorithms deliver robust data products (e.g. Spreen et al. (2008); Markus and Cavalieri (2000)) at the one to ten kilometre scale. Observation of processes at finer spatial scales can only be carried out by active sensors. One such instrument capable of higher resolution observations is synthetic aperture radar (SAR), delivering year-round backscatter measurements that are sensitive to changes in the ice cover. Due to the diverse backscattering properties that sea ice admits it's diverse development from frazil to perennial ice (see, e.g. Onstott (1992); Kortum et al. (2022, 2024)) the corresponding data is





more difficult to interpret than optical satellite imagery. This complex relationship between radar backscatter and the physical
state of sea ice is a central complication for retrieval algorithms. An alternative approach to high resolution monitoring of sea
ice is the use of altimeters, which detect the distance to the ground in nadir. In the case of the laser altimeter on ICESat-2,
footprint sizes of the measurement are on the order of tens of meters as detailed in Neumann et al. (2019). Altimeter measure-
ments have low uncertainties of only a few centimeters in their height retrievals and thus allow precise measurements of the

distance between the satellite and the scatterer on the ground. If leads open in the ice cover up and the open water is detected,
this distance can be used as reference for the sea surface height. Measurements of the surrounding sea ice surface then can be
converted to a freeboard measurement, as described in Kwok et al. (2022). This is the total height of the ice and snow above
the sea surface. Not only is the freeboard indicative of the ice development, series of such measurements can be combined
into a topographic understanding of the surface, describing roughnesses at various scales. A large blind spot of the altimetry

measurement is given by its spatially sparsity in the transversal/across-track direction of the flight path, as measurement takes
place only along thin lines over the Arctic. Tracks from multiple flights can be combined to give a large-scale overview on a
monthly basis. However, resulting gridded products (Petty et al. (2020)) are constrained to a more regional scale (25 km grid
cell size) and have to be aggregated for about one month to achieve pan-Arctic coverage.

SAR and altimetry data both yield valuable insights into the Arctic system. At the same time they are complementary in

a variety of aspects: SAR has large 2-dimensional coverage, whilst altimetry coverage is sparse. However, converting radar
backscatter data into key measurements of the sea ice is very challenging, whilst laser altimetry measures the sea ice height
very precisely, is easy to interpret and gives concrete information about the sea ice topography. Because of that some research
already exists concerning the combination of both instruments. Karvonen et al. (2022) combined Sentinel-1 SAR and CryoSat-
2 radar altimeter measurements of ice thickness, seeking to levarage the advantages of each technique. The technique they

developed uses the SAR data to interpolate between the altimetry data at kilometer scale, by segmenting the SAR image and
assigning CryoSat-2 measured ice thicknesses to segments. Recently Macdonald et al. (2024) published a study over landfast
ice in the Canadian Arctic Archipelago, in which correlations of altimeter measurements (roughness, freeboard) and C-Band
SAR HV backscatter appeared stronger than those with HH backscatter. Their research also suggested that a roughness re-
trieval from SAR HV data is feasible. Concerning roughness and SAR HH/VV backscatter, strong correlations ($R_p$ = 0.82)

where found by Cafarella et al. (2019) under shallow incidence angles for first-year ice and similar correlations ($R_p$ = 0.74)
where observed in Segal et al. (2020) also over the Canadian Arctic Archipelago. Meaningful correlations of surface roughness
at smaller scales could not be observed in Kortum et al. (2024) for X-Band SAR and LiDAR data, with ($R_{Pearson}$ < 0.3) over
mixed, multiyear ice, second-year ice and first-year ice over a small area of sea ice in the central Arctic. In this work we
present correlations of freeboard and roughness with C-Band SAR at a near pan-Arctic scope and demonstrate an algorithm to

55 extrapolate ICESat-2 altimetry-derived freeboard to Sentinel-1 SAR scenes at up to 100 m resolution.

In this study we are not proposing that SAR backscatter is a direct indicator of sea ice thickness (which might be question-
able). We are only using the backscatter intensity in the vicinity of actual ICESat-2 (Ice, Cloud and Land Elevation Satellite)
ice freeboard measurements to extrapolate them in space. Locally one can expect that the ice thickness to SAR backscatter
relationship is stable enough to retrieve sea ice freeboard for the whole SAR scene. We extrapolate ICESat-2 freeboard heights





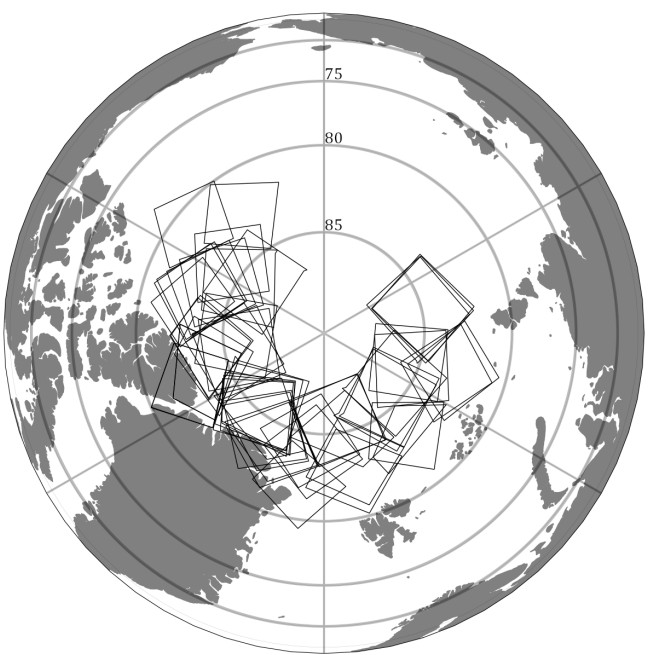

**Figure 1.** Location of all Sentinel-1 scenes from October or November (2018-2022) with near coincident ICESat-2 coverage. These acquisitions are the main source of data for this study.

to coincident Sentinel-1 SAR scenes, which were acquired within plus/minus 24 hours of the ICESat-2 overflight. This enables observations near the spatial scope and frequency of the Sentinel-1 constellation, which is considerably larger than the altimeter coverage alone, but the errors are higher than for the altimetry data, because of the limited correlation of sea ice backscatter and freeboard. A freeboard product at spatial resolution of up to 100 m and time intervals and coverage of the Sentinel-1 mission, as proposed here, is none-the-less a useful resource for polar research and stakeholders.

## 2 Data

An overview over all data products that are used in this study is given below.

The first source data product we use, are Sentinel-1 SAR acquisitions, captured in EW (Extended Wide) mode. Scenes captured in this acquisition mode, have a footprint of approximately 400 km by 400 km with an individual pixel size of 40 metres. We use the Ground Range Detected (GRD) product, which projects the measurement to geo coordinates using an earth ellipsoid model. The terrain correction in the Sentinel-1 Toolbox in SNAP (SNAP (2022)) is used to correct these measurements with a geoid model, which is close to the ocean height and reduces the geolocation error. The incidence angle range of the scenes is between 20 and 50 degrees. Thermal noise, scalloping and calibration to $\sigma_0$ is done using the SNAP (2022) library and corrections developed by the Nansen Center and described in Park et al. (2018, 2019); Korosov et al. (2022). These mitigation




measures help minimise the effect of sensor artefacts on the study. To allow for more ICESat-2 footprints to fit into one pixel, and thus to derive more meaningful statistics, the scenes are then resampled to 100 x 100 metre pixel spacing. This also mitigates speckle effects. The footprints of all scenes used in this study are plotted in figure 1 for an overview.

On the altimetry side, we are using ICESat-2's ATL-10 sea ice freeboard measurement. ICESat-2 is an optical laser altimeter that operates at a wavelength of 532 nm and is highly pulsed at 10.000 pulses a second. The resulting altimetry measurement is accurate to approximately 2 cm. Because the freeboard segments are dependent on the scattering conditions of the surface (a certain number of photons is collected per segment), the ATL-10 product's spacing is variable and on the scale of tens of metres. At these intervals segments are returned with a freeboard height and expected variance. To have as many data points as possible, we use the three weak beams as well as the three strong ones, giving us a maximum of 6 beams from which data can be used. Due to atmospheric conditions and the requirement of nearby open leads a freeboard measurement is not always available when the instrument is measuring.

The bulk data in the study consists of 59 Sentinel-1 EW scenes, along with all ICESat-2 ATL-10 freeboard data within 24 hours of the SAR aqcuisition over the same footprint. The specific SAR scenes are selected, because there exists an ICESat-2 overflight that is near coincident with the SAR measurement (time difference is less than 10 minutes) and the ATL-10 freeboard tracks overlap with at least 300 pixels ($100 \times 100\,\mathrm{m}^2$) of the SAR scene. In fact, these 59 scenes are all EW acquisitions between 2018 and 2022 in October and November, that had a near-coincident acquisition of ICESat-2. The near-coincident flights are important to observe the correlations between the measurements and later on to validate the extrapolation results. October and November are selected, because of two reasons: Firstly, there exist comparatively many near-coincident acquisitions in this time period. This is likely due to atmospheric conditions, i.e., less clouds, as ICESat-2's laser at 532 nm does not penetrate these. Secondly, first-year ice is still quite young at this point and can therefore be more easily be distinguished from older ice in both SAR and altimetry missions. As a result the correlations between freeboard and backscatter are expected to be highest during this time of the Arctic sea ice cycle.

Setting the maximum time difference for a 'near-coincident' measurement at 10 minutes and with pixel sizes of 100 metres, significant decorrelation of both measurements can start to occur if the ice drifts faster than 50 metres in 10 minutes (= 300 m/h). Such high drift speeds are reached occasionally, but this constraint to 10 minutes is sufficient to make sure the vast majority of data points are still valuable. The data are matched using the geocoding of both products used and no ice drift correction is applied. For Sentinel-1 geolocation uncertainties are reported by Schubert et al. (2017) to be around 5m over land, which we can use as a baseline error. Additionally the geoid model used for the ground range projection will have an error relative to the real sea surface height, that should be of a similar scale as the local sea surface height anomaly. Skourup et al. (2017) investigated the model and observational differences and found differences in the central arctic up to 0.5 m. Thus we can assume the Sentinel-1 geocoding error to be generally below 10 m. ICESat-2 geolocation errors are reported to be around 2.5 m to 4.4 m by Luthcke et al. (2021). With pixel sizes of 100 m being significantly larger than the uncertainties of geocoding, this should be sufficient to get meaningful overlap between the SAR and freeboard products at this scale.

All ICESat-2 ATL-10 segments in one Sentinel-1 pixel are considered equally: To obtain a local freeboard the mean of all freeboard segment heights from ATL-10 pertaining to a pixel is taken. For roughness we investigate two different considera-



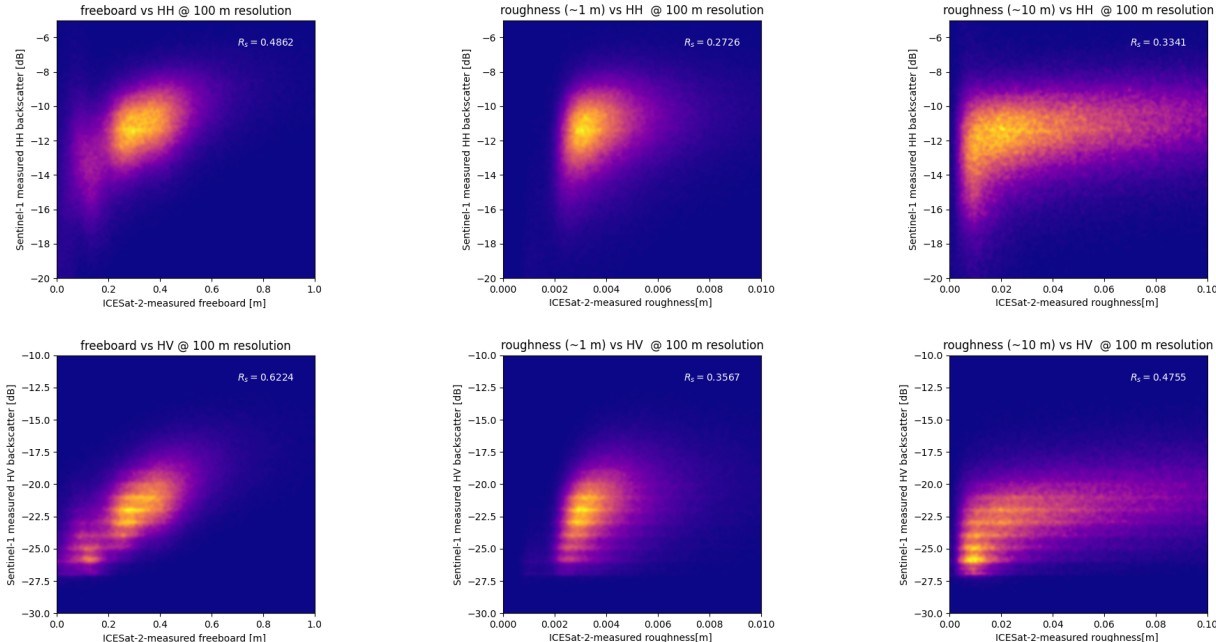

**Figure 2.** 2D Histograms of ICESat-2 and Sentinel-1 measurements and the respective Spearman correlation coefficients. Brighter colours correspond to higher data density, whilst darker blueish colors correspond to lower density. Some banding effects are visible in the HV channel.

tions, that describe different roughness correlation lengths (scales). The ATL-10 product gives an expected variance for each freeboard segment, determined by local photon statistics and thus approximately at meter scale. For the first roughness observation, all freeboard segments' respective variances are summed up and from the square root of their mean a final sigma value for each pixel is obtained giving a roughness at approximately the meter scale. Alternatively a larger scale roughness can be obtained by calculating the standard deviation of all ATL-10 freeboard segment heights within one 100 m x 100 m SAR pixel. The correlation lentght/scale of this roughness measure is equivalent to the spacing of the segments, i.e. on the order of 10s of meters. Both of these roughness measures use the variance of freeboard heights as a proxy for roughness of the ice surface.

## 2.1 Correlations

We will first investigate the statistical connections between the altimetry and SAR data. In this case we are mainly interested in the correlations of these variables, as that will be of importance for the extrapolation measures described later.

Heatmaps of both the freeboard and the roughness from all 597,565 data points are plotted in figure 2, along with the respective Spearman correlations. We use Spearman correlations here, as we do not expect the values to be correlated linearly, but we are interested in how accurately we could construct a monotonic mapping from one to the other - which is exactly what the Spearman correlation coefficient captures. In table 1 the Spearman correlation coefficients are listed. The split into





multiyear (MYI) and first-year (FYI) ice is performed for 51 of these scenes (with 392,364 data points), which admitted a clearly bimodal freeboard distribution, allowing to differentiate between the two ice types via thresholding the freeboard.

There are three main studies from the Canadian Arctic Archipelago we can compare these results to, all of which focus on fast ice. Cafarella et al. (2019) investigated the statistical relationship of high resolution C- and L-band SAR data (resolution≈ 10 and 3 m, respectively) with LiDAR derived sea ice roughness (resolution= 1.2 m) over first year ice. From two scenes acquired in the late winter season (March, April), they found a high correlation (Pearsons R) of 0.86 for high incidence angles (46 deg) and low correlation of 0.30 for low incidence angles, for the HH backscatter and roughness. The correlation of the

HV backscatter and roughness was found to be more similar across the two scenes at around 0.81 for high and 0.68 for low incidence angles. Segal et al. (2020) observed the correlations of LiDAR derived roughness, a roughness proxy from the MISR optical satellite and Sentinel-1 C-Band SAR over first-year and multiyear ice in late winter (April). The roughness was derived from 1 m resolution LiDAR data and the grid cells were 1.2 km by 0.4 km large. They found a high correlation (Perasons's) for roughness and HH backscatter at 0.74 across their whole dataset, with 0.76 on only first-year and 0.12 on only multiyear ice.

Recently, Macdonald et al. (2024) published a study comparing SAR and altimetry measurements for three ICESat-2 flights in the Canadian Arctic archipelago in March. It is also worth noting that they computed roughnesses from the University of Maryland supersampled ICESat-2 product, described in Duncan and Farrell (2022); Farrell et al. (2020). As the source for SAR data they used the Radarsat Constellation Mission (RCM) in a low noise mode unique to the instrument and found (Spearman) corrrelations for first-year ice roughness and SAR backscatter at 0.42 for the HV and 0.31 for the HH channel. The correlations

with mutli-year sea ice height and backscatter were 0.49 in the HV and 0.41 in the HH channel. They also demonstrated an accurate roughness retrieval at 800 m scale. The differences of these previous studies and ours are the spatial scales, seasons and location. While these previous studies were looking at a more regional scale, we have gathered more flights from more diverse Arctic regions. However, our roughness measures are not as fine-scale or accurate as the LiDAR data or the University of Maryland product. Additionally we are focussing on the early rather than the late winter season.

The freeboard correlations with the HV channel across our entire dataset are remarkably strong at $0.62$. The correlation for MYI and HV channel is the same as in the Macdonald et al. (2024) study at $0.49$. The correlations with the roughness are, however, weaker, especially in the HV channel than in all previous studies. Causes for this could be the ice development, because of the difference in ice seasons or the roughness measures used. Comparably low correlations were also found in Kortum et al. (2024) for sea ice roughness at length scales of 0.5 m with the HH and VV channels of X-band SAR. The

correlations for freeboard might be stronger in this study in contrast with the Macdonald study, because of the rescaling to 100 m x 100 m, that should lead to an increase of correlations as quasi-random speckle effects average out. Additionally, the study area and time might be a cause for this, with both very thin first year ice and the oldest, thickest perennial ice being captured within this studies' dataset. This should also lead to an increase in correlation.





**Table 1.** Spearman correlations coefficients of ICESat-2 and Sentinel-1 measurements. The correlations for HH and HV are calculated from all 59 available flights. Of these 51 admitted a bimodal freeboard distribution, allowing the separation of first-year ice (FYI) and multiyear ice (MYI).

|  | HH | HH(FYI) | HH(MYI) | HV | HV(FYI) | HV(MYI) |
|---|---|---|---|---|---|---|
| freeboard | 0.49 | 0.18 | 0.34 | 0.62 | 0.32 | 0.49 |
| roughness (1 m) | 0.27 | 0.24 | 0.20 | 0.36 | 0.26 | 0.31 |
| roughness (10 m) | 0.33 | 0.19 | 0.22 | 0.48 | 0.36 | 0.36 |

## 3  Methods

### 3.1  Algorithm Structure

The structure of the proposed freeboard extrapolation method using SAR backscatter is as follows.

1. For the SAR scene to be used as basis for the extrapolation, all ATL-10 measurements within the last 24 hours are retrieved.

2. A mapping is constructed from the HV SAR data to non-coincident measured ATL-10 freeboard in the area via the cumulative distribution functions of the HV SAR measurement and the altimeter freeboard product.

3. The mapping is applied to the HV channel of the entire scene from step 1.

This extrapolation using the cumulative distribution functions entirely relies on the correlations of sea ice ageing processes and it's freeboard, illustrated in figure 3. As young ice freezes up, a brine expulsion on top of the ice leads to wet and saline surface and possibly wetted snow, as investigated by e.g. Drinkwater and Crocker (1988). This lossy material is quite absorbent and backscatter is typically quite low, especially for double bounces required for HV returns. Whatever backscatter is measured probably originates from surface roughness features, which also increase freeboard. As the ice gets older and desalinates (Cox and Weeks (1974)), the penetration of the radar measurement increases and volume scattering from bubbles and empty brine channels begins to increase the HV signal. Finally large topographical features such as ridges can accommodate double bounce backscatter returns and can again increase the HV backscatter return. It is important to keep in mind, however, there is no direct physical connection between the backscatter and ice freeboard. I.e. there is no physical reason why a ridge 1.5 m high should have a stronger HV backscatter response than one only 1 m high and this is therefore the strongest limitation of this approach. We, however, propose that in the vicinity of a measured freeboard distribution from ICESat-2, the backscatter is a reasonable predictor of relative freeboard heights and can therefore be used to extrapolate the freeboard measurements. This is possible, becaise the freeboard distribution in the majority of cases does not change drastically on a 100-km scale and within 24 hours.



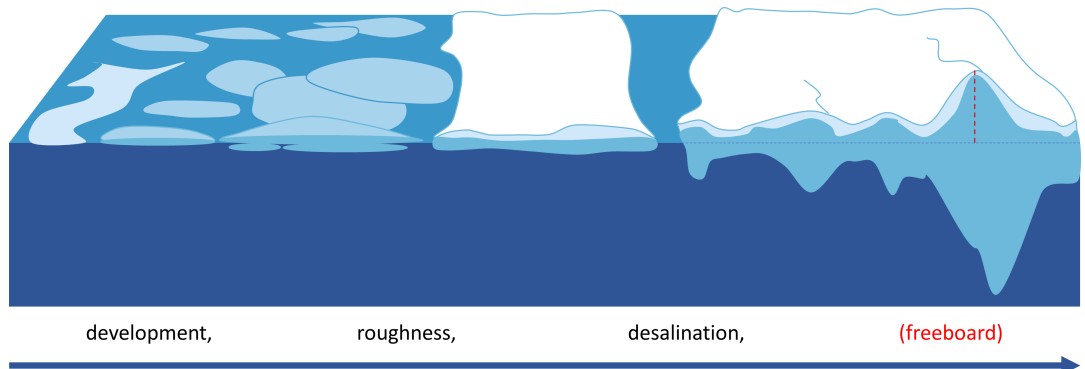

**Figure 3.** Illustration of the connection between freeboard and ice development responsible for the increase in HV backscatter (mainly desalination and surface roughness increase).

## 3.2 Cumulative distribution function (CDF) mapping

To create the mapping between ICESat-2 freeboard and Sentinel-1 backscatter via the cumulative distribution functions (CDFs) all ATL-10 data from the last 24 hours within the boundaries of the SAR scene are collected. Their cumulative distribution function $\mathrm{CDF}_{fb}$ is formed from all measurements taken. For the CDF of the HV channel $\mathrm{CDF}_{HV}$, all pixels within 1000 m of an ICESat-2 track are considered. Because the ice has drifted in between the measurements, it is not the exact same ice forming both CDFs, however restriction to the approximate area does ensure that the distribution of the underlying ice is similar. The constructed map via the CDFs is illustrated in figure 4 and can be expressed as

$$
\begin{aligned}
&\Phi : \{\sigma_{HV}\} \mapsto \{fb\} \\
&\Phi(\sigma_{HV}) = (\mathrm{CDF}_{fb}^{-1} \circ \mathrm{CDF}_{HV})(\sigma_{HV})
\end{aligned}
\tag{1}
$$

With this mapping constructed, pixels can be mapped from HV backscatter to freeboard for the entirety of the Sentinel-1 acquisition. It is worth noting that the Spearman correlation coefficient is invariant under such a monotonic transformation. Thus all the improvement between the Spearman correlations of the predicted freeboard and the measured freeboard in contrast to the HV backscatter and the measured freeboard comes from the different CDF mappings for each scene.

## 3.3 Validation

To validate the results of the method, the procedure as described above is carried out for all 59 scenes in the dataset. To form the cumulative distribution function $\mathrm{CDF}_{fb}$ for the freeboard, all ATL-10 data within 24 hours of the acquisition are taken, except the ATL-10 validation flight within ten minutes. Then the extrapolated freeboard is compared with the near-coincident validation flight over the same scene. That way the comparisons with the validation flights are carried out in an unbiased manner, independent of the constructed mapping and therefore representative of the algorithm performance.





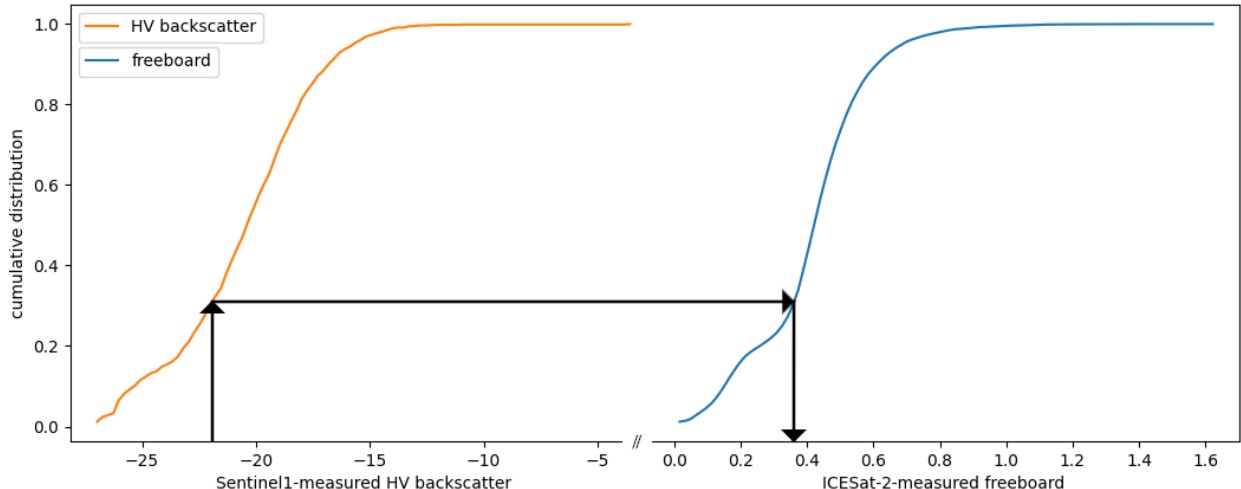

**Figure 4.** Visualisation of the mapping constructed from the cumulative distribution functions of freeboard and HV backscatter. The black path illustrates a mapping from an HV backscatter value to a freeboard value.

## 4  Results

Figure 5 shows the central results of the predicted algorithm. At 100 m resolution a Pearson correlation of $0.68$ between
measured and extrapolated freeboard, shows that relationship of HV backscatter and freeboard can be used to make meaningful
extrapolation possible. The errors, however, are still considerable at just above 10 cm. Judging also by the heatmap in Figure
5, at 100 m resolution this technique enables the separation of ice into approximate classes such as first-year or multiyear ice
and to detect ridges. As the resolution is lowered, the retrieval method becomes increasingly accurate, as is illustrated by the
narrowing of the heatmap. At 400 m resolution, with Pearson correlation $R_p = 0.82$ and errors of 6 cm, the retrieval method
shows promising results that can unlock comprehensive freeboard surveys of the Arctic in 2 dimensions.

An example scene is shown in figure 6, where qualitatively the extrapolated freeboard aligns well with the overlayed ATL-
10 measurements. The bottom track is shown in more detail below in Figure 7, where it becomes clear that in most cases
the characteristics are captured well ($R_{\text{Pearson}}$=0.67), but the exact height (especially of ridged areas) cannot be approximated
accurately (RMSE=0.08).Occasionally some younger ice areas are shown to be significantly thinner than assumed from the
extrapolation. These areas have also posed problems in sea ice classification algorithms in the past, as described for example
in Guo et al. (2022).



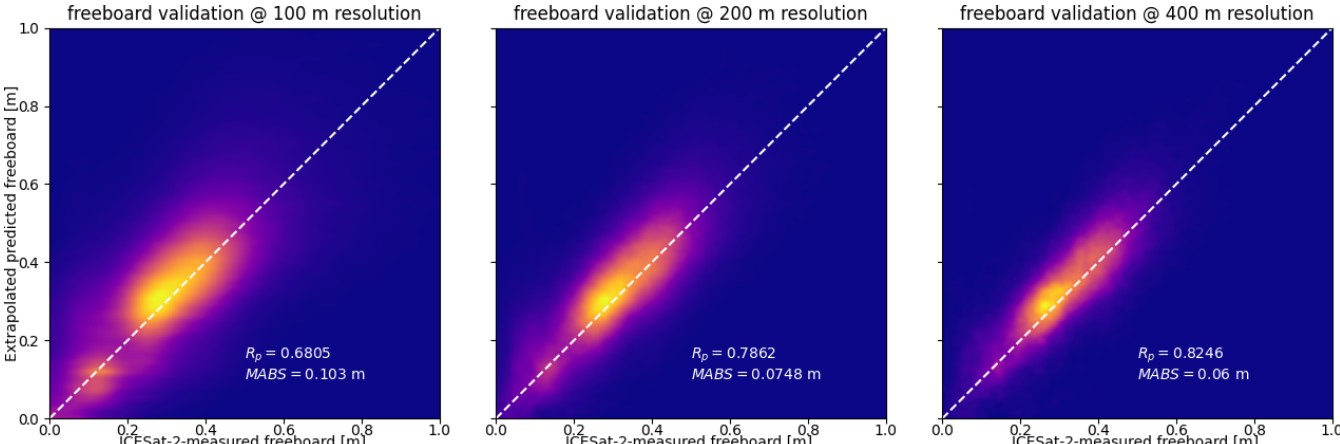

**Figure 5.** Results of the extrapolated freeboard data on all 59 scenes (597,565) data points, with Pearson correlation coefficients $R_p$ and mean absolute errors (MABS) shown in the figures. Brighter areas indicate a higher density.

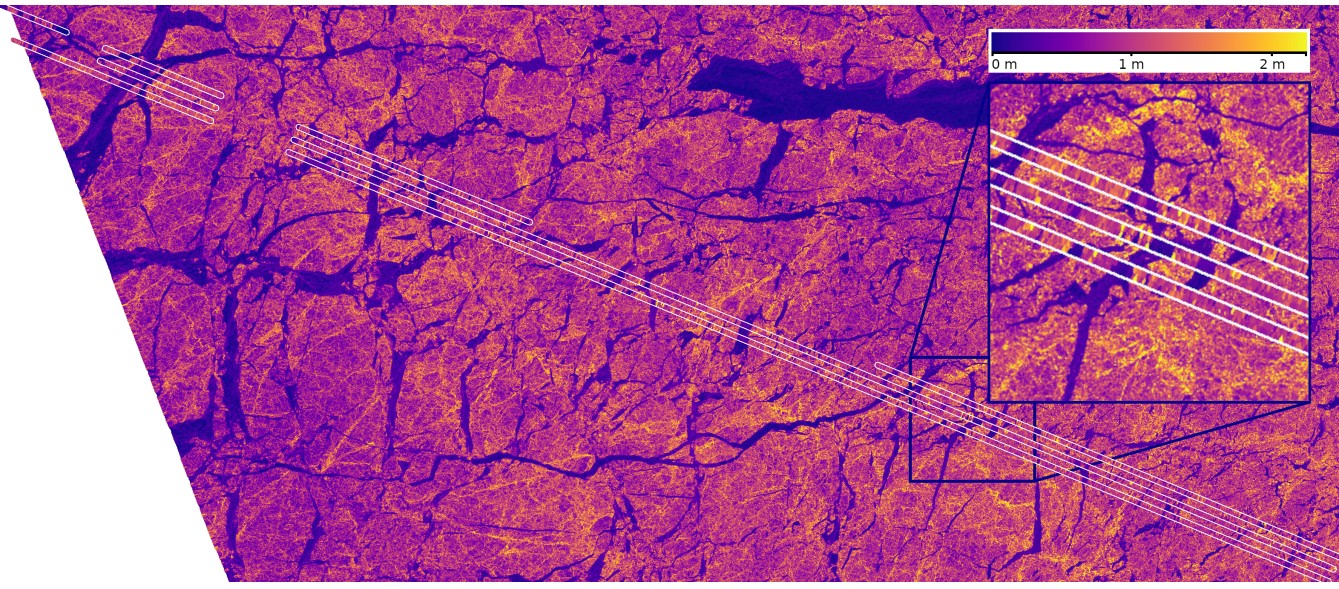

**Figure 6.** Example scene from the 29th of November 2021 with both the extrapolated freeboard at 100 m resolution and overlayed ICESat-2 ATL-10 data. The ATL-10 data were thickened artificially (using nearest neighbour extrapolation) to allow easier visualisation and are shown within the white contour. The three visible tracks are made up of one strong and one weak beam each.



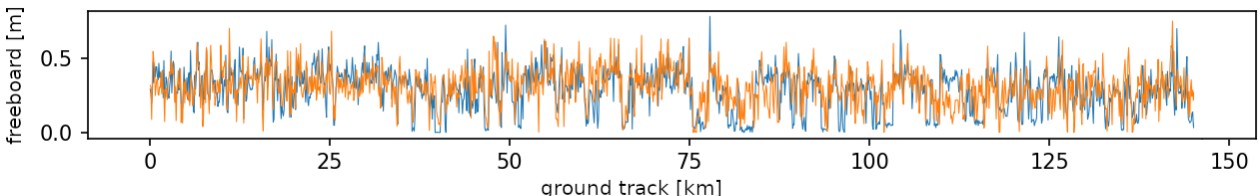

**Figure 7.** Bottom track from figure 6 with measured (ICESat-2, blue) and extrapolated (Sentinel-1, orange) freeboard values at 100 m spacing.

## 5 Discussion

The correlations between the SAR backscatter and altimetry freeboard and roughness data in this dataset largely differ from the ones observed in previous studies by Cafarella et al. (2019); Segal et al. (2020); Macdonald et al. (2024). As mentioned earlier, the study area and time are probably the main reasons for this. From our observations it seems that relating roughness and backscatter is more difficult in the early winter season focussed on in this work, than in the late winter seasons investigated by the previous studies. However, correlations with freeboard are still significant, which reinforce the notion that they can successfully be related to one another. This study is the first time these correlations could be observed for drifting sea ice across a large area in the Arctic and the correlations of freeboard and the HV channel are remarkably high, considering that there is no direct physical connection between backscatter and freeboard.

The results reveal that the proposed algorithm enables meaningful extrapolation of ice freeboard as measured by ICESat-2, capturing the key features and revealing the spatial variability of freeboard in two dimensions at 100 m to 400 m resolution and for the coverage of Sentinel-1 scenes. The accuracy of the retrieval is difficult to judge in relation to other methods as no comparable products exist. The algorithm performs accurate enough to separate ice types and ridges at 100 m resolution with errors around 10 cm and could be useful as a direct approximation of freeboard at 400 m resolution given an error of approximately 6 cm.

As previously mentioned, the main source of the remaining retrieval uncertainties is the limitation of physical connection between topography and SAR backscatter - something that cannot be circumvented. Additional sources of error do also exist, however. For example the footprints of ICESat-2 are not covering the entire pixel they are being mapped to, meaning the ground truth we use for freeboard in every pixel is already contaminated by this undersampling. Next to the existing uncertainty of the ATL-10 products, SAR noise and speckle effects also are additional sources of error. Additionally the overlap of the validation flights is limited by the accuracy of the georeferencing of the sensors. In the case of Sentinel-1, the GRD product uses an ellpsoid model which can vary up to 10s of meters from the real ocean surface height.

So far, the extrapolation has been limited to only a certain season in the year, i.e. October/November, where older and younger ice have significantly different freeboards, which increase the correlation with SAR backscatter. Expanding this approach to other seasons and the marginal ice zone will be more challenging. Part of the reason is, that the amount of overlap-





ping data at 10 minutes of time difference, needed to validate the results, is sparser in other months and non-existing inside the marginal ice zone.

We have validated the approach with independent, near coincident ICESat-2 flights. Comparison with CryoSat-2 radar
altimeter measurements would be the next logical step. Because of the different dominant scattering surface of that radar instrument, however, the freeboard measured by CryoSat-2 is different than that measured by ICESat-2, as shown in Fredensborg Hansen et al. (2024) using the Cryo2Ice data. Therefore it is less useful as validation data. It would be very interesting, however, to investigate the possibility of extrapolating CryoSat-2 and future CRISTAL measurements using the same method and comparing the results. Additionally, the new SWOT altimeter allows for 2D freeboard retrieval that would be a great can-
didate for validation, or extrapolation. However, the coverage is restricted to 78° North/South, and therefore it's use for sea ice application is unfortunately quite limited, but a case-study based comparison might be possible.

For future work, it might also be possible to increase the accuracy of the extrapolation with advanced statistical or machine learning methods. However, the initial ideas that we attempted yielded no significant improvement.

Whilst we worked with extrapolating ICESat-2s ATL-10 product from NASA, other current or future altimetry products
might also be able to be extrapolated with SAR. For example, the previously mentioned University of Maryland product by Farrell et al. (2020); Duncan and Farrell (2022) would be worth using instead of the ATL-10 data for roughness approximation as was done in Macdonald et al. (2024). As the main focus was shifted to freeboard in this study, this was not considered.

The uses of a medium to high resolution freeboard product are manifold. The data can be used as a good proxy to sea ice thickness in terms of variability in two dimensions, something that has so far alluded consistent observation. Maritime
stakeholders might also profit from these data, as well as weather and climate models, the former of which could be initialised with observations in near real time. High resolution digital twin earth models, as are currently in development by Hoffmann et al. (2023) at ECMWF would especially benefit from these observations, due to their km-scale grid spacing.

## 6   Conclusions

Our work presented in this manuscript shows how ICESat-2 derived freeboard measurements can be meaningfully extrapolated
with Sentinel-1 SAR measurements at resolutions up to 100 m for the entire 400km SAR scene with up to a 24 hour time difference between the SAR and altimetry acquisitions. This algorithm opens up an opportunity to monitor Arctic wide sea ice freeboard in two dimensions, capturing its spatial variability at previously unattainable coverage and making in important step towards monitoring ice thickness. It is yet to be shown that this approach can also work throughout all seasons and regions in the Arctic.

*Acknowledgements.* This study was funded by Deutsche Forschungsgemeinschaft (DFG) under project name 'MOSAiCmicrowaveRS' (SI 2564/1-1 and SP 1128/8-1).



We would like to thank all people involved at ESA's Copernicus program for the acquisition and provision of Sentinel-1 SAR data. Additionally we express our thanks to NASA and everyone involved there with the acquisition and distribution of ICESat-2 data products.

*Author contributions.* Karl Kortum - Conceptualisation, Formal Analysis, Investigation, Methodology, Writing - original draft.
Suman Singha - Funding acquisition, Project administration, Supervision, Writing - review and editing.
Gunnar Spreen - Funding acqiusition, Supervision, Writing - review and editing.

*Competing interests.* The authors have no competing interests to declare.



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
