# Peer review of "Sea Ice Freeboard Extrapolation from ICESat-2 to Sentinel-1"

_EGUsphere, 2024_

## Referee Comment (RC1)

**Sea Ice Freeboard Extrapolation from ICESat-2 to Sentinel-1**

Karl Kortum, Suman Singha2, and Gunnar Spreen

**12 Jan 2025**

**General comments**

The authors present a study aimed at extrapolating along-track ICESat-2 sea ice freeboard heights to two-dimensional Sentinel-1 freeboard heights by using cumulative distribution functions (CDFs). The study utilizes 59 Sentinel-1 scenes co-located with ICESat-2 ATL10 data, examining the correlation between freeboard heights (roughness) and Sentinel-1 HH/HV backscatter. Guided by these correlations, the authors employ CDFs to map ICESat-2 data onto Sentinel-1 imagery. The extrapolated freeboard estimates are then compared with the near-coincident validation track over the same scenes.

Overall, the work is promising. As the authors highlight, achieving two-dimensional sea ice freeboard measurements could significantly enhance the capacity to monitor sea ice at spatial scales that were previously difficult to attain. However, before the manuscript can be accepted, the authors should address the following concerns and make the method more convincing:

1. Using CDFs to link radar backscatter (especially HV polarization) to the distribution of sea ice freeboard has a certain degree of validity. However, questions remain about the method's underlying assumptions. CDF matching aligns distributions based solely on the observed data, without necessarily establishing a direct physical link between HV backscatter and freeboard. Consequently, when applying the extrapolation, high HV backscatter is mapped to high freeboard, while low HV backscatter is mapped to low freeboard. Given the complexity of sea ice conditions, such a straightforward relationship may be insufficient. Radar backscatter can be influenced by multiple factors, including salinity, surface roughness, and snow depth. Sea ice with identical freeboard heights might exhibit different backscatter signals, potentially leading to inaccuracies in freeboard

estimation when relying on such a simplified mapping strategy.

2. The manuscript mentions 59 datasets. It appears that a separate CDF mapping is created for each dataset, rather than one universal mapping for all 59 datasets. If this is correct, it implies that the method cannot be directly applied to new Sentinel-1 data for which no coincident ICESat-2 data exist. This limitation would significantly reduce the broader applicability of the technique. Clarifying whether a single CDF mapping was generated or multiple mappings were used—and if the latter, how the authors envision applying the method to future acquisitions—would be helpful.

3. The study uses ICESat-2 data acquired within 24 hours of the corresponding Sentinel-1 data. However, some of the validation data were obtained under near-coincident conditions (e.g., time differences of less than 10 minutes). Given the rapid drift of sea ice, non-coincident ICESat-2 observations may not perfectly align with the Sentinel-1 pixels, thus introducing potential errors in correlations. It would be beneficial to explain why strictly near-coincident data (e.g., with a time difference of less than 10 minutes) were not used to establish the CDF mappings directly.

4. I guess that using both freeboard height from strong beams and weak beams is not a better way to calculate the mean freeboard height in a pixel. Generally, weak beam segments are typically about four times longer than strong beam segments, the resulting freeboard estimates from weak beams may be smoother. It would be more informative to analyze and present the correlations separately for strong and weak beams, thereby highlighting potential differences or biases in the derived freeboard estimates.

**specific comments**

line 94: 'be distinguished' should be 'distinguished';

Line 175: 'becaise' should be 'because';

Figure 6: suggest that plot the raw Sentinel-1 image as well to compare.

---

## Referee Comment (RC2)

The paper introduces an approach for extrapolating sea ice freeboard provided by ICESat-2 altimeter laser data to the larger area covered by a nearly coincident Sentinel-1 SAR scene at a high 100 m resolution. The algorithm arises from a relatively high correlation between HV backscatter and the ICESat-2 freeboard. The accuracy of the approach was assessed from a comparison against independent ICESat-2 data acquired within 10 minutes of SAR acquisitions. The paper can be considered for publication after the following comments are fully addressed.

Major comments:

1. The paper lacks a comparison of the algorithm against a completely independent (of ICESat-2) in-situ source of ice freeboard data. One possible in-situ data source could be long-term time-series records of upper looking sonar data in the Beaufort Sea, in particular Mooring B, that is located above 78 degree north. The authors may suggest an alternative in-situ source of ice freeboard data, but I believe that an assessment against independent in-situ data is absolutely necessary to demonstrate the usefulness of the proposed approach.

2. The algorithm is based on HV backscatter, that is much lower than HH, so HV is substantially affected by the thermal noise. Noise floor in Sentinel-1 EW data is very high (e.g., compared to the noise floor in RCM data); additionally, the noise exhibits a scalloping pattern in the image azimuth direction. The authors applied a noise correction routine in SNAP, but while this correction makes the HV image look nicer (less affected by noise), it does not really make HV signal more informative (i.e., it does not increase the HV signal dynamic range over darker targets like first-year sea ice). The different noise floor levels in different SAR instruments could substantially affect the correlations/relationships between HV and the freeboard/roughness. Therefore, the authors should investigate how the relatively high noise floor in Sentinel-1 affects their freeboard/roughness retrievals. This is especially important for thinner first-year ice where the HV signal could be very low reaching the noise floor level. I also wonder if it is feasible to build and assess the algorithm without the noise correction, but with using noise floor information as auxiliary input piece of information.

3. HV backscatter is sensitive to the incidence angle. The authors should discuss how the incidence angle variation in the image (20-50 degrees) affects the accuracy of the freeboard retrieval.

Minor comments:

Overall, the manuscript contains a lot of typos and inaccuracies, and some of them, yet not all, are outlined below. It feels that the manuscript preparation was rushed. The authors should carefully correct all inconsistencies and typos.

Line 72. "Thermal noise, scalloping and calibration to σ0 is done …". It is not clear what it exactly means. Does this mean that thermal noise and scalloping effect reductions were applied? Please rephrase.

Line 75. "100 x 100 metre" -> "100 m x 100 m"

Lines 76, 119, 163, 181 and throughout the text where appropriate. "figure" -> "Fig."

Figure 2. The quality of the figure should be improved. Add the single color bar for all the panels. Add space between roughness and [m] in x-axis. All panels show rather backscatter vs freeboard/roughness and not vice versa. Add letters to each panel.

Line 93. "be distinguished" -> "distinguished"

Line 100. "5m" -> "5 m"

Line 122. "table 1" -> "Table 1".

Line 131. "C-Band" -> "C-band".

Line 150. "Macdonald"->"Macdonald et al. (2024)"

Line 167. "penetration of the radar measurement" does not sound correct

Line 170. "freeboard. I.e." -> "freeboard, i.e.,"

Line 174. "becaise" -> "because"

Figure 5. Color bar should be added.

Line 204. Spacing is missing in ".Occasionally".

Line 228. "ellpsoid" -> "ellipsoid"

Line 255. "400km" -> "400 km"

Line 276. "Requencies" -> "Frequencies"

---

## Author Comment (AC1)

**Reviewer 2 Response**

Thank you for the comments and suggestions. We will follow up on the feedback and improve/clarify the manuscript accordingly.

1. *Using CDFs to link radar backscatter (especially HV polarization) to the distribution of sea ice freeboard has a certain degree of validity. However, questions remain about the method's underlying assumptions. CDF matching aligns distributions based solely on the observed data, without necessarily establishing a direct physical link between HV backscatter and freeboard. Consequently, when applying the extrapolation, high HV backscatter is mapped to high freeboard, while low HV backscatter is mapped to low freeboard. Given the complexity of sea ice conditions, such a straightforward relationship may be insufficient. Radar backscatter can be influenced by multiple factors, including salinity, surface roughness, and snow depth. Sea ice with identical freeboard heights might exhibit different backscatter signals, potentially leading to inaccuracies in freeboard estimation when relying on such a simplified mapping strategy.*

   We agree that the direct linkage of HV backscatter can only have a limited, and entirely statistical, relationship with freeboard and we have tried to be clear about the limitations in the manuscript - pointing this out multiple times. We believe that despite the limitations of this CDF mapping approach it is still an important baseline method for extrapolation. The advantage of this approach lies its simplicity, making it easy to understand the results. As there is further research currently being conducted into these topics, it is especially important to have a simple approach to compare future methods with, as they should be as least as accurate as the simple CDF-based extrapolation scheme presented here.
   We will add some of these thoughts and usefulness of the model for comparison with more advanced future methods into the discussion.

2. *The manuscript mentions 59 datasets. It appears that a separate CDF mapping is created for each dataset, rather than one universal mapping for all 59 datasets. If this is correct, it implies that the method cannot be directly applied to new Sentinel- 1 data for which no coincident ICESat-2 data exist. This limitation would significantly reduce the broader applicability of the technique. Clarifying whether a single CDF mapping was generated or multiple mappings were used—and if the latter, how the authors envision applying the method to future acquisitions—would be helpful.*

   You are correct that a mapping is constructed for each scene. However, it is not constructed from the coincident ICESat-2 data. Instead it is constructed from ICESat-2 from data within 24 hours of the SAR acquisition. As we are observing close to the poles, sufficient ATL-10 data to construct the map is usually available for scenes in the observed season (freeze-up) and this approach can be applied to most Sentinel-1 scenes.
   We will add some additional clarification on the paper to shore up this point.

3. *The study uses ICESat-2 data acquired within 24 hours of the corresponding Sentinel-1 data. However, some of the validation data were obtained under near-coincident conditions (e.g., time differences of less than 10 minutes). Given the rapid drift of sea ice, non-coincident ICESat-2 observations may not perfectly align with the*

*Sentinel-1 pixels, thus introducing potential errors in correlations. It would be beneficial to explain why strictly near-coincident data (e.g., with a time difference of less than 10 minutes) were not used to establish the CDF mappings directly.*

This goes along well with the comment/answer above. You are correct, that the validation data is all from near-coincident conditions. The non-coincidence of extrapolated data certainly introduces an error, that arises from the non-matching. However, accepting this error allows this method to be broadly applicable. Otherwise its use would be constrained to very rare near-coincident passes of both satellites. We will make sure to emphasise this in the manuscript in the next iteration.

4. *I guess that using both freeboard height from strong beams and weak beams is not a better way to calculate the mean freeboard height in a pixel. Generally, weak beam segments are typically about four times longer than strong beam segments, the resulting freeboard estimates from weak beams may be smoother. It would be more informative to analyze and present the correlations separately for strong and weak beams, thereby highlighting potential differences or biases in the derived freeboard estimates.*

We actually tried using strong beams only, but saw (slight) improvement from including weak beam data as well. We currently expect there to be little difference between using weak and strong beams, as the uncertainties are probably smaller than those arising from the limited physical connection of freeboard and backscatter. We will investigate again with only weak and only strong beams and report the results in the next version of the manuscript.

Thank you also for the specific comments, we will implement these suggested changes.

---

## Author Comment (AC2)

**Reviewer 1 Response**

Thank you for your comments and suggestions. Each of them will be addressed individually in the following and respective changes will be made to the manuscript.

Major Comments.:

1. *The paper lacks a comparison of the algorithm against a completely independent (of ICESat-2) in-situ source of ice freeboard data. One possible in-situ data source could be long-term time-series records of upper looking sonar data in the Beaufort Sea, in particular Mooring B, that is located above 78 degree north. The authors may suggest an alternative in-situ source of ice freeboard data, but I believe that an assessment against independent in-situ data is absolutely necessary to demonstrate the usefulness of the proposed approach.*

   Thank you for the idea of comparing to upward looking sonar data. We agree that an additional comparison with entirely independent data would be a good addition to the manuscript. Unfortunately, there is no available above snow freeboard for the investigated time period, that could be used as validation. However, a comparison with upward looking sonar would be a good demonstration of the type of capabilities that an extrapolated freeboard product provides. We will work on including such a comparison in the next version of the manuscript.
   We also want to set the expectations of such a comparison. The question we ask in the paper currently is "how well can we extrapolate ICESat-2 data using Sentinel-1?". The only way to answer that question is to compare the extrapolated data with real ICESat-2 data. From the coincident flights we have gathered a dataset consisting of approximately 500.000 datapoints. We believe this dataset is comprehensive enough to answer the question raised (for sea ice conditions similar to the ones investigated). The additional comparison with ULS will not give us a qualitative assessment of the extrapolation algorithms accuracy, because of the limited physical connection between ice draft and above snow freeboard. However, it will show how two measurement that could previously not be collocated can be brought together. We also expect that the same trends should be visible in both measurements.

2. *The algorithm is based on HV backscatter, that is much lower than HH, so HV is substantially affected by the thermal noise. Noise floor in Sentinel-1 EW data is very high (e.g., compared to the noise floor in RCM data); additionally, the noise exhibits a scalloping pattern in the image azimuth direction. The authors applied a noise correction routine in SNAP, but while this correction makes the HV image look nicer (less affected by noise), it does not really make HV signal more informative (i.e., it does not increase the HV signal dynamic range over darker targets like first-year sea ice). The different noise floor levels in different SAR instruments could substantially affect the correlations/relationships between HV and the freeboard/roughness. Therefore, the authors should investigate how the relatively high noise floor in Sentinel-1 affects their freeboard/roughness retrievals. This is especially important for thinner first-year ice where the HV signal could be very low reaching the noise floor level. I also wonder if it is feasible to build and assess the algorithm without the noise correction, but with using noise floor information as auxiliary input piece of information.*

   Thank you for your suggestion. We applied the thermal noise removal by Park, Korosov et al, (see Data section, L. 73) which does help with the overall brightness

and therefore the accuracy of the extrapolation. You are right that locally no information is gained in areas with higher noise floors and therefore the extrapolation there is expected to perform worse. Currently I see no way of incorporating the noise floor information to sharpen the CDF-mapping.

To investigate the contribution of this noise to the errors is a fruitful idea. We will conduct an ablation study where the noisiest parts of the Sentinel-1 imagery are not used for the extrapolation to determine the impact of SAR signal noise and report the results.

3. *HV backscatter is sensitive to the incidence angle. The authors should discuss how the incidence angle variation in the image (20-50 degrees) affects the accuracy of the freeboard*

While HV backscatter is sensitive to the incidence angle, the dependencies are quite small (compared to HH). So, we do not expect there to be a large impact. We will include some additional discussion in the revised version and try to quantify the influence of this effect on the accuracy of extrapolation.

Minor Comments:

Thank you for finding and pointing out these errors. We will do our best to clean up the spelling and formatting for the rest of the document too.

---

## Referee Report (RR1)

**Sea Ice Freeboard Extrapolation from ICESat-2 to Sentinel-1**

Karl Kortum, Suman Singha, and Gunnar Spreen

**July 14, 2025**

**General comments**

As a first-round reviewer, I would like to thank the authors for their efforts to incorporate the reviewers' comments and enhance the manuscript. I see the quality of the manuscript has been improved significantly. However, some typos and inaccuracies were not still handled well. I have provided several specific comments below, and I encourage the authors to pay more attention to addressing these. The authors should check the manuscript carefully before publication.

**specific comments**

- 1. Line 234. "30dB" -> "30 dB".
- 2. Table 2 & Figure 5. "mean absolute error (MABS)". I don't think "MABS" is the abbreviation for "mean absolute error". "MAE" should be correct.
- 3. Line 135. "Pearsons R" -> "Pearson's R".